# Catalytic CO$_2$ Desorption Study of Tri-Solvent MEA-EAE-DEEA with Five Solid Acid Catalysts

Huancong Shi [1,2,*] , Yingli Ge [1], Shijian Lu [3,*], Jiacheng Peng [1], Jing Jin [1] and Liangquan Jia [4]

1    School of Energy and Power Engineering, Department of Environmental Science and Technology, University of Shanghai for Science and Technology, Shanghai 200093, China
2    Huzhou Institute of Zhejiang University, Huzhou 313000, China
3    Carbon Neutrality Institute, China University of Mining and Technology, Xuzhou 221008, China
4    School of Information Engineering, Huzhou University, Huzhou 313000, China
*    Correspondence: hcshi@usst.edu.cn (H.S.); lushijian@cumt.edu.cn (S.L.)

**Abstract:** To achieve the CO$_2$ emission control as the urgent task of Carbon Peak and Carbon Neutrality, the CO$_2$ desorption experiments were performed with a new tri-solvent MEA-EAE(2-(ethylamino)ethanol)-DEEA(N, N-diethylethanolamine) with five solid acid catalysts: blended catalysts of γ-Al$_2$O$_3$/H-ZSM-5 = 2:1, H-Beta (Hβ), H-mordenite, HND-8, and HND-580 as H$_2$SO$_4$ replacement. A series of sets of experiments were performed in a typical recirculation process by means of both heating directly at 363 K and temperature programming method within 303~358 K to evaluate the key parameters: average desorption rate (ADR), heat duty (HD), and desorption factors (DF). After analyses, the 0.5 + 2 + 2 mol/L MEA-EAE-DEEA with catalyst HND-580 possessed the best CO$_2$ desorption act at relatively low amine regeneration temperatures with minimized HD and the biggest DF among the other catalysts. Comparing with other tri-solvents + catalysts studied, the order of DF was MEA-BEA-DEEA + HND-8 > MEA-EAE-DEEA + HND-580 ≈ MEA-EAE-DEEA + HND-8 > MEA-EAE-AMP + HND-8. This combination has its own advantage of big cyclic capacity and wider operation region of CO$_2$ loading range of lean and rich amine solution ($\alpha_{lean}$~$\alpha_{rich}$), which is applicable in an industrial amine scrubbing process of a pilot plant in carbon capture.

**Keywords:** catalytic CO$_2$ desorption; amine regeneration; tri-solvent; heat duty; desorption factor

## 1. Introduction

CO$_2$ capture, utilization, and storage technology (CCUS) were important and applicable technologies to reach the target of the Paris Agreement, and "Carbon peaks 2030" and "Carbon Neutrality 2060" in P. R. China [1–3]. The post-combustion carbon capture (PCCC) technology was one of the almost industrial implemented techniques in power plants, the steel industry, cement plants, etc., to mitigate CO$_2$ emissions. The massive energy cost of CO$_2$ desorption is the major bottleneck, which contributes to up to 70% of the overall energy cost. This drawback prohibits the commercialization of PCCC technology, and it is urgent to adopt energy-saving methods as useful solutions [1–3]. From the previous research, solvent improvement [3,4] and heterogeneous catalysis [5,6] were two useful approaches since 2000. Meanwhile, the PSA was a strong potential candidate technique [7–9].

Many researchers reported the advantage of amine blends as improved solvents. Most of them were bi-blends, and some studies reported tri-blends in recent years. Bi-blends have been investigated and studied intensively for more than two decades and were categorized into five to six small branches [10,11]. Tri-blends, labeled as amine A+B+C, have started to raise research interests since 2016 [11–23]. Most tri-blends were prepared with a similar methodology: combining primary or secondary amines with tertiary amines (R$_3$N) or stereo-hindered amines (AMP), such as MEA-PZ-AMP [18–21] and MEA-MDEA-PZ [16,17]. The selection of MEA or PZ fully adopted their superior advantage with fast

absorption rates. The blending of AMP or MDEA was intended to reduce heat duty based on their advantage of desorption performance [16].

The optimized blending ratio of Amine A+B+C needs to be figured out by experiments so that the optimized tri-solvents can be used in a pilot plant to fit the $CO_2$ absorption-desorption conditions with balance, which delivers a suitable operation region ($\alpha_{lean} \sim \alpha_{rich}$) and cyclic capacity for absorbent [24,25]. However, the methodology of blending tri-solvents of A+B+C at different blending ratios was quite complicated than bi-solvents of A+B. The total amine concentration $C_A$ cannot be too small with reduced cyclic capacity (CC) or too big to crystalize and precipitate [24]. Based on a large number of experiences, $C_A$ should be larger than 4 mol/L and smaller than 7 mol/L, which was more suitable between 4.5~6.0 mol/L [16,24,25].

A small literature review of recently published tri-solvents was categorized in 2022 already [12], which reported: MEA-MDEA-PZ [16,17], MEA-AMP-PZ [18–21], DETA-AMP-MDEA [22,23], MEA-BEA-AMP [11,15], and MEA-BEA-DEEA [14], etc. After 2020, the energy-efficient tri-solvents had drawn strong research interests, which were: MEA-BEA-DEEA [14], MEA-EAE-AMP [12,13]. Such studies effectively revealed several energy-saving tri-solvents much better than 5.0 a mol/L MEA as benchmark.

Besides tri-solvent, heterogeneous catalytic $CO_2$ desorption is another useful energy-saving tactic since 2010 [5,11,14,26–39]. A special review of 2020 [5] reported the comprehensive analysis of catalytic $CO_2$ desorption of MEA solvent with a large number of heterogeneous catalysts, focusing on heterogeneous catalysis, characterization of catalysts in surface area and pore volume/pore size, structure-activity, correlations, and detailed mechanisms on a molecular level [5]. The solid acid catalysts were highly energy efficient for $CO_2$ desorption, which can significantly cut the heat duty and reduce $CO_2$ desorption temperatures to the range of 95–98 °C, which is under the boiling point of water with reduced latent heat.

Since 2021, the authors focused on the mixture of tri-solvent with the aid of heterogeneous catalysts and completed several publications which were MEA-BEA-AMP + blended $\gamma$-$Al_2O_3$/H-ZSM-5 = 2:1 [11], MEA-BEA-DEEA + HND-8 [14], MEA-EAE-AMP + HND-8 [12], MEA-MDEA-PZ + H$\beta$ [16], MEA-AMP-PZ + HZSM-5 [18], etc. However, the tri-solvent of MEA-EAE-DEEA has not been studied yet. Compared with BEA, EAE possessed better $CO_2$ absorption performance, with a wider range of operation ($\alpha_{lean} \sim \alpha_{rich}$) [24]. Compared with AMP, DEEA has a lower energy cost of desorption and a larger cyclic capacity [24]. The MEA-EAE-DEEA tri-solvents might possess moderate desorption behavior between MEA-EAE-AMP and MEA-BEA-DEEA. The heat duty (HD), cyclic capacity (CC), and desorption factors (DF) of the tri-solvent + catalysts were well worth studying; as well, the mixture of MEA-EAE-DEEA with solid acid catalysts has not been published yet, to the best of our knowledge.

The $CO_2$ desorption experiments were preformed onto tri-solvent MEA-EAE-DEEA for this study at amine concentrations ($C_A$) of 0.4 + 2 + 2 and 0.5 + 2 + 2 mol/L by means of temperature programming with several solid acid catalysts: H$\beta$, H-mordenite, HND-580, and HND-8. The catalytic $CO_2$ desorption performance was investigated systematically, with typical parameters such as: heat duties (HD), average desorption rates (ADR), and desorption factors (DF) [12].

The purpose of this study: (1) Investigate the $CO_2$ desorption performance of tri-solvents of "MEA-EAE-DEEA" with five solid acid catalysts to expose the most energy-saving combination. (2) The heat duty (HD) and desorption factor (DF) of various MEA-EAE-DEEA amine solutions with various catalysts were analyzed to compare with MEA-BEA-DEEA and MEA-EAE-AMP with the same sets of solid acid catalysts.

## 2. Theory

### 2.1. Mechanism: The Coordinative Effects Existing in MEA and RR'NH (EAE)

The "coordinative effect" had been published within MEA + RR'NH bi-solvents, which were MEA-DEA [39], MEA-BEA [40], MEA-EAE [13] and tri-blends of MEA-BEA-AMP [15] with MEA-EAE-AMP [13]. This effect was published to exist in MEA-EAE already [13]. The heat duty of bi-blends MEA-EAE at a specific ratio was lower than that of EAE alone.

The optimized ratio of MEA/EAE was 0.4/2 for $CO_2$ desorption within MEA-EAE-AMP at 0.2 + 1 + 3, and 0.3 + 1.5 + 2.5 mol/L [12].

This phenomenon of MEA-RR'NH having better desorption performance than RR'NH alone seems to be opposite to the well-acknowledged concept that MEA boosts $CO_2$ absorption but sabotages desorption in blended amine solutions [24]. The intrinsic mechanism of this effect works is quite unique: blending 0.1~0.5 mol/L MEA raises heat input $Q_{input}$ to 5–10% (negative to HD), while it enhanced $nCO_2$ production to 10–20% (positive to HD). This simultaneous effect of increasing both $Q_{input}$ and $nCO_2$ comprehensively reduces the overall heat duty (HD = $Q_{input}$/$nCO_2$) down to 5–10% [13,40]. The coordinative effect belongs to the latter, and a detailed mechanism had been published repeatedly [10,13,15,39–42].

### 2.2. The Mechanism of Catalytic CO₂ Desorption

This mechanism was originally proposed by the author Shi et al. in 2011 [43,44]. Later on, the review published in 2020 organized several mechanisms of catalytic $CO_2$ desorption [5], which consisted of several key steps in order: (1) "carbamate formation", (2) "carry protons", (3) "chemical adsorption", (4) "isomerization", (5) "stretching", (6) "C-N bond cleavage/breaking", and final step of (7) "desorption/separation". This mechanism was described in detail by various types of catalysts with multiple schemes, which were published by various research groups and categorized in 2020 [5]. The carbamate of MEA and RR'NH were easy to break down or hydrolysis with solid acid catalysts introduced into solvents, which could be proceeded below 100 °C. Therefore, the heat duty of $CO_2$ desorption was significantly reduced with the neglected latent heat of steam [5].

### 2.3. The Average Desorption Rate, Heat Duty, and Desorption Factor of CO₂ Desorption Analysis

The average $CO_2$ desorption rates (ADR) were calculated in Equation (1) [35]. The heat duty (HD) was a critical parameter of desorption performance evaluation, and calculated with $Q_{input}$/$nCO_2$ in Equation (2) [14]. The heat input ($Q_{input}$) was tested with an electrometer, and $CO_2$ production ($nCO_2$) was estimated with ($\alpha_{rich} - \alpha_{lean}$) × C × V [11,26]. The electrometer recorded the data at different testing time period of 0, 30, 60, 120 and 180 min and the unit was converted from kW·h to kJ (1.0 kW·h = 3600 kJ). The calculated HD was much bigger than the industrial value of 3.5–4.0 GJ/t$CO_2$ in a steady-state pilot plant because of different calculation equations toward different experimental apparatuses. This study adopted a batch scale process with relatively big heat loss. However, such phenomena occurred in all cases, so that the HD of this study was still comparable to each other. Recently, $CO_2$ desorption performance was evaluated with the desorption factor [45].

$$\text{Average desorption rate} = \frac{nCO_2}{\text{time}} \tag{1}$$

$$\text{HD} = \frac{\text{Heat input/time}}{nCO_2/\text{time}} = \frac{\text{Electricity (kJ)}}{nCO_2 \text{ (mol)}} \tag{2}$$

$$\text{Desorption Factor} = \frac{\text{Average Desorption Rate} \times \text{Cyclic Capacity}}{\text{Heat Duty}} \tag{3}$$

$$\text{Cyclic Capacity} = C_A \left( \alpha_{rich} - \alpha_{lean} \right) \tag{4}$$

### 3. Results and Discussion

This work contains three sets of: (1) tri-solvent MEA-EAE-DEEA (0.1 + 2 + 2~0.5 + 2 + 2 mol/L) with blended acid catalysts of $\gamma$-$Al_2O_3$/HZSM-5 = 2:1 to discover the optimized blending ratio, and then compare optimized amine concentration with max $C_A$ + optimized catalysis. (2) MEA-EAE-DEEA (0.4 + 2 + 2 mol/L and 0.5 + 2 + 2 mol/L) with the rest 4 solid acid catalysts to discover an energy-efficient combo. The DF of tri-solvents + catalysts were analyzed to screen out a suitable combo for the industrial amine scrubbing process in pilot

plants scale. (3) The heat duty HD, and desorption factor DF were compared with two other sets of MEA-EAE-AMP and MEA-BEA-DEEA + catalysts through comprehensive consideration.

### 3.1. Catalytic $CO_2$ Desorption of MEA-EAE-DEEA with Blended Solid Catalysts of $\gamma$-$Al_2O_3$/HZSM-5 with Direct Heating

The catalytic $CO_2$ desorption tests were performed within MEA-EAE-DEEA with blended catalysts of $\gamma$-$Al_2O_3$/HZSM-5 = 2:1. The blended solid acid catalysts consist of both Br$\phi$nsted acid and Lewis acid sites. The order of catalysis was reported as blended catalysts $\gamma$-$Al_2O_3$/HZSM-5 = 2:1 > HZSM-5 > $\gamma$-$Al_2O_3$ [11,14,26,29,39]. The series of studies were intended to compare the optimized blending ratio with moderate catalysis vs. max $C_A$ or cyclic capacity with optimized catalysis.

### 3.1.1. The Non-Catalytic and Catalytic $CO_2$ Desorption of MEA-EAE-DEEA

Figure 1a–e plotted the non-catalytic and catalytic $CO_2$ desorption profiles were plotted of MEA-EAE-DEEA at 0.1 + 2 + 2~0.5 + 2 + 2 mol/L. The $C_A$ of EAE and DEEA was fixed at 2.0 mol/L to facilitate absorption and desorption, and the ratio of MEA/EAE ranged within 0.1~0.5/2 to find out the optimized coordinative effect [12]. For each set of tri-solvents with various $C_A$, catalytic desorption curves were below than their non-catalytic curves, indicating the effectiveness of solid acid catalysts [26].

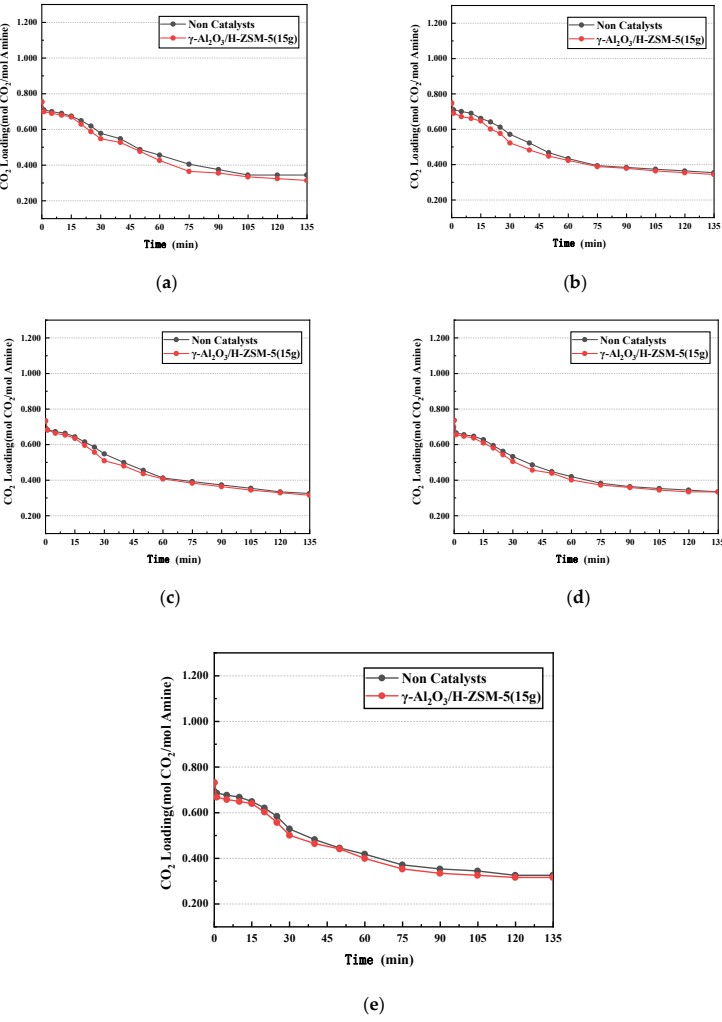

**Figure 1.** The $CO_2$ desorption curves of MEA-EAE-DEEA with blended solid catalysts from 0.1 + 2 + 2~0.5 + 2 + 2 mol/L. (**a**) 0.1 + 2 + 2 mol/L, (**b**) 0.2 + 2 + 2 mol/L, (**c**) 0.3 + 2 + 2 mol/L, (**d**) 0.4 + 2 + 2 mol/L, (**e**) 0.5 + 2 + 2 mol/L.

Figure 2a,b plotted the heat duties (HD) of the tri-solvents for 30 and 60 min. The HD were calculated in Equation (2) based on the $CO_2$ loading $\alpha$ in the desorption curves of Figure 1. It was discovered the HD of MEA-EAE-DEEA with catalytic $CO_2$ desorption was smaller than its non-catalytic counterparts. As a consequence, catalytic $CO_2$ desorption rates were higher than non-catalytic benchmarks.

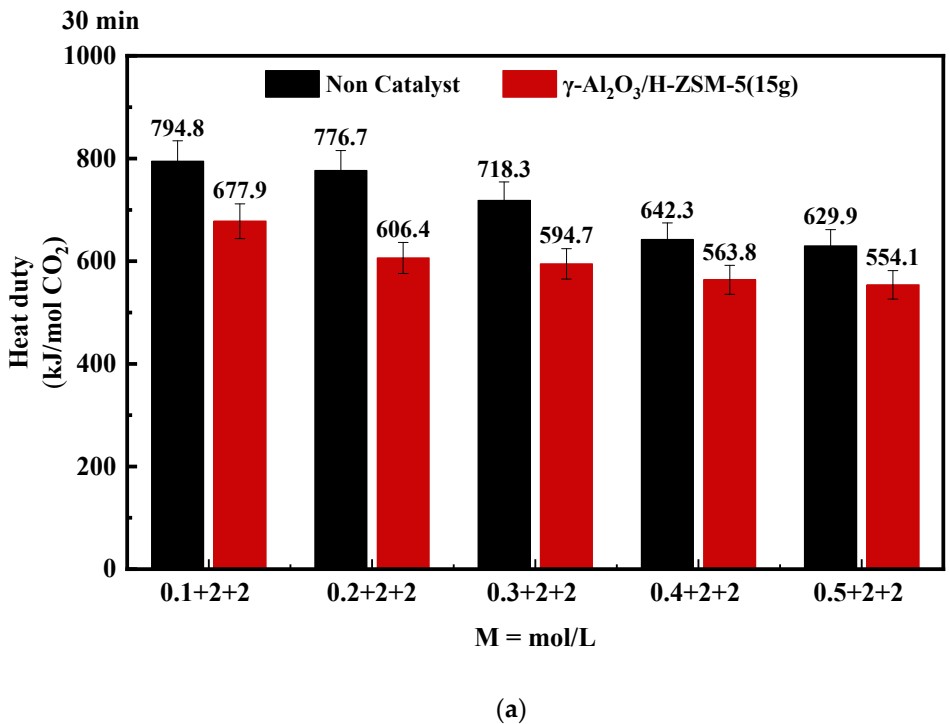

(a)

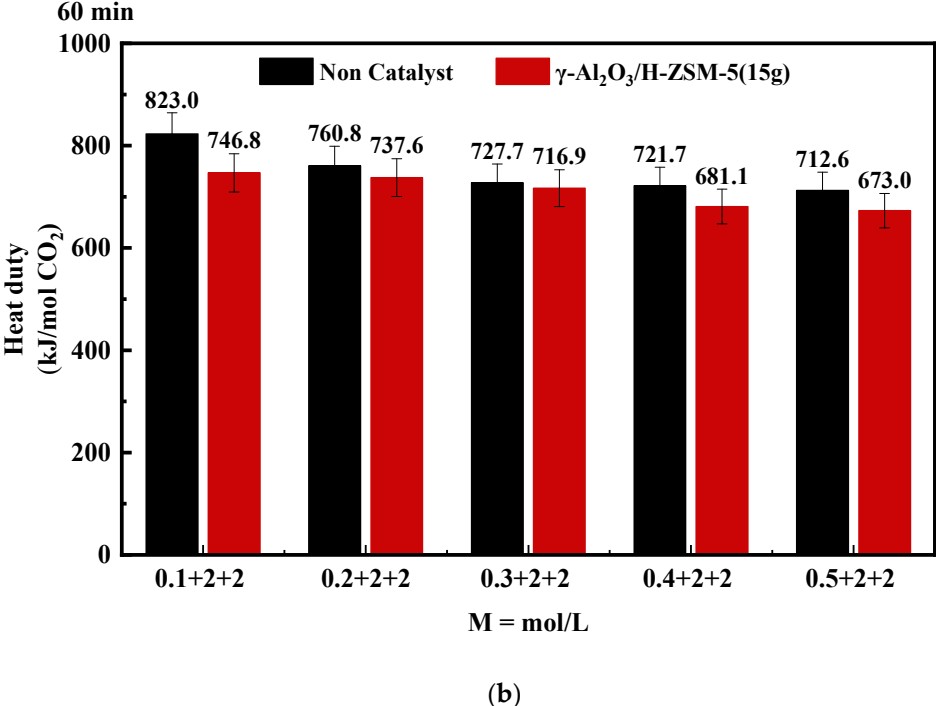

(b)

**Figure 2.** The heat duties (HD) of MEA-EAE-DEEA with blended acid catalysts at (**a**) 30 and (**b**) 60 min.

Later on, Figure 3a,b plotted the average desorption rates (ADR) at 30 and 60 min based on the slopes of the desorption curve of Figure 1. Figures 1–3 reflected repeatedly the

effectiveness of solid acid catalysis, which can boost $CO_2$ desorption [5]. The role of solid acid catalysts helps to facilitate N-C bond cleavage of carbamate, accelerate $CO_2$ desorption rates, emissions of $CO_2$ out of amine solvent, and then boost ADR as well as increase $nCO_2$ and reduce HD [5].

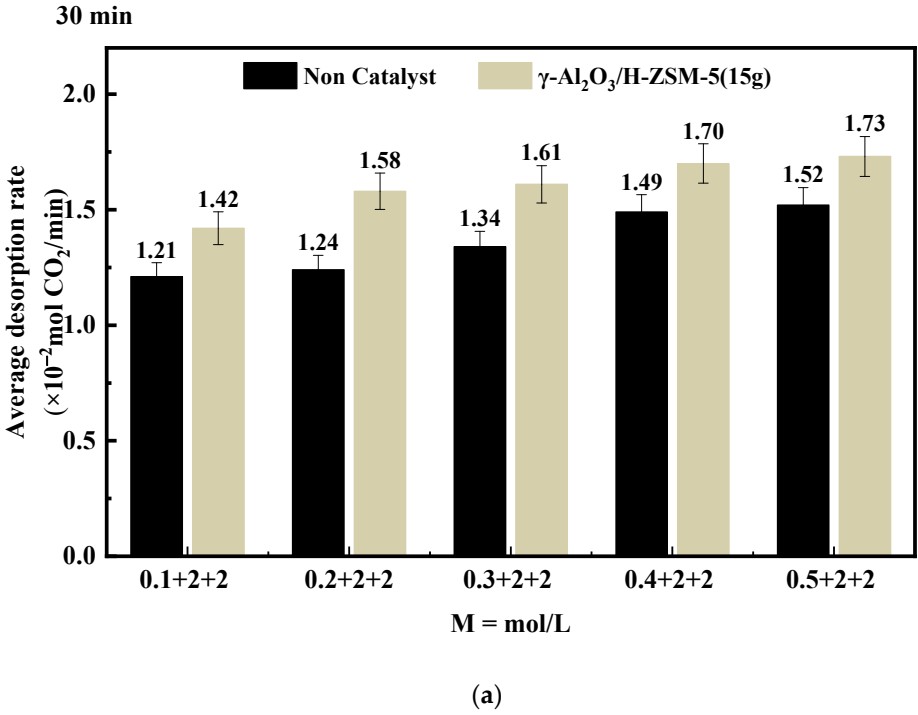

(**a**)

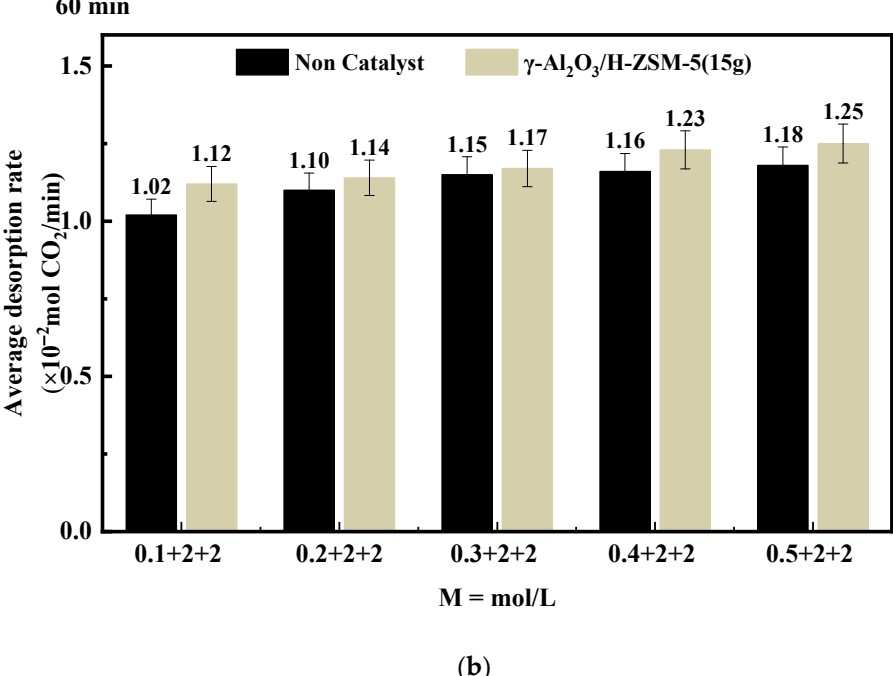

(**b**)

**Figure 3.** The average desorption rates (ARD) of MEA-EAE-DEEA with solid acid catalysts: (**a**) 30 min; (**b**) 60 min.

Based on Figures 1–3, the optimized blending ratio of MEA/EAE was 0.4~0.5/2 with minimum heat duty. In previous publication, the coordinative effect in the tri-solvent were resulted from MEA and EAE(RR′NH) only [13], since MEA-DEEA($R_3N$) contained

negligible coordination [40]. The DEEA ($R_3N$) could not produce any carbamate, so that the mechanism of coordinative effect does not exist [40]. The optimized ratio of 0.5/2 MEA/EAE was comparable to the results of a previous study of 0.4/2 of MEA/EAE within MEA-EAE-AMP [13]. From Figure 2, the HD of the tri-solvent of 0.4/2/2 was very close to 0.5/2/2 with a difference less than 2.5%. If considering experimental error of 3%, both 0.4 + 2 + 2 mol/L and 0.5 + 2 + 2 mol/L were regarded as the optimized blending ratio.

From Figure 2, the tendencies were consistent under both catalytic and non-catalytic cases: the HD of tri-solvents decreased sharply and reached the minimum value at the edge condition of 0.5 + 2 + 2 mol/L. For catalytic desorption, the HD of 0.4 + 2 + 2 was only 1.7% and 1.1% higher than that of 0.5 + 2 + 2 mol/L at both 30 and 60 min. Therefore, the optimized amine concentration was within 0.4 + 2 + 2~0.5 + 2 + 2 mol/L, and 0.5 + 2 + 2 mol/L were considered optimized candidates based on current experimental results.

The DF were categorized into Table 1 to compare different combinations of tri-solvent + solid acid catalysts under a consistent level. From Table 1, the optimized mixture of tri-solvent with $\gamma$-$Al_2O_3$/HZSM-5 was 0.5 + 2 + 2 mol/L based on the biggest DF.

**Table 1.** The desorption factors of tri-solvent of MEA-EAE-DEEA solvents with $\gamma$-$Al_2O_3$/H-ZSM-5 at 30 min and 60 min.

| MEA-EAE-DEEA | Desorption Factor ($10^{-3}$ mol $CO_2$)$^3$/$L^2$ kJ min | | | |
|---|---|---|---|---|
| | 30 min | | 60 min | |
| (mol/L) | No Catalyst | $\gamma$-$Al_2O_3$/H-ZSM-5 (15 g) | No Catalyst | $\gamma$-$Al_2O_3$/H-ZSM-5 (15 g) |
| 0.1 + 2 + 2 | 0.0110 | 0.0178 | 0.0152 | 0.0203 |
| 0.2 + 2 + 2 | 0.0118 | 0.0248 | 0.0192 | 0.0211 |
| 0.3 + 2 + 2 | 0.0149 | 0.0263 | 0.0220 | 0.0230 |
| 0.4 + 2 + 2 | 0.0209 | 0.0309 | 0.0225 | 0.0268 |
| 0.5 + 2 + 2 | 0.0221 | 0.0325 | 0.0234 | 0.0278 |

### 3.1.2. Comparison with Tri-Blends of MEA-EAE-AMP and MEA-BEA-DEEA + $\gamma$-$Al_2O_3$/HZSM-5 Catalyst

The results were compared with MEA-EAE-AMP + $\gamma$-$Al_2O_3$/HZSM-5 catalyst based on a recent publication [12], with similar operation conditions, amine concentrations, and catalysts. Therefore, the comparisons can evaluate difference catalysis on a consistent level.

The HD of four combinations were categorized into Table 2, indicating both optimized and edge concentrations. The HD of catalytic $CO_2$ desorption of MEA-EAE-AMP at 30 min was relatively lower than that of MEA-EAE-DEEA, and the range of HD was 533.7~590.8 kJ/mol which is lower than that of 554.1~677.9 kJ/mol [12]. The optimized catalysis was 0.2 + 2 + 2 mol/L MEA-EAE-AMP with HD of 533.7 kJ/mol, and the boundary condition was 0.5 + 2 + 2 mol/L with HD of 557.4 kJ/mol [12]. Figure 2a reported optimized condition of 0.5 + 2 + 2 mol/L MEA-EAE-DEEA with HD of 554.1 kJ/mol. These results indicated the optimized mixture of MEA-EAE-AMP + solid acid catalyst was 21 kJ/mol (4%) lower than that of MEA-EAE-DEEA with the aid of blended $\gamma$-$Al_2O_3$/HZSM-5 = 2:1, while the boundary condition of MEA-EAE-AMP was only within 1% different with of MEA-EAE-DEEA.

The HD of MEA-EAE-AMP vs. MEA-EAE-DEEA was comparable to each other with MEA-EAE-AMP slightly lower since both AMP and DEEA were both energy efficient solvent amines. A previous study compared MEA-BEA-AMP vs. MEA-BEA-DEEA under similar operation conditions, the absolute heat duty (HD) of MEA-BEA-DEEA was slightly smaller than the HD of MEA-BEA-AMP under 30 min [14]. Such a difference may be due to different secondary amines of EAE vs. BEA. The exception was MEA-EAE-DEEA, where the boundary concentration was the same as the optimized concentration. Based on Table 2, the effects of AMP and DEEA were close and comparable to each other, with slight differences of heat duty. The overall HD of MEA-BEA-AMP/DEEA were much lower, indicating the much better desorption performance of BEA vs. EAE.

**Table 2.** The HD of 4 different tri-solvents with $\gamma$-Al$_2$O$_3$/HZSM-5 at minimum and edge conditions at 30 min.

| Tri-Solvents | Concentration (mol/L) | Heat Duty (kJ/mol CO$_2$) | Ref. |
|---|---|---|---|
| MEA-EAE-AMP | 0.2 + 2 + 2 mol/L | 533.7 | [12] |
| | 0.5 + 2 + 2 mol/L | 557.4 | [12] |
| MEA-EAE-DEEA | 0.5 + 2 + 2 mol/L | 554.1 | This study |
| MEA-BEA-AMP | 0.3 + 2 + 2 mol/L | 144.7 | [11] |
| | 0.5 + 2 + 2 mol/L | 157.0 | [11] |
| MEA-BEA-DEEA [a] | 0.3 + 2 + 2 mol/L (15min) | 100.7 | [14] |
| | 0.5 + 2 + 2 mol/L (15 min) | 103.0 | [14] |

[a] The HD of MEA-BEA-DEEA was very small, which report 0–15 min only.

### 3.2. Catalytic CO$_2$ Desorption of Tri-Solvents MEA-EAE-DEEA with 4 Commercial Catalysts

Since both tri-solvents were selected as energy-saving approaches, it is reasonable to conduct a thorough analysis of 0.4 + 2 + 2 mol/L and 0.5 + 2 + 2 mol/L with several commercial solid acid catalysts. The difference in HD of both tri-solvents was less than 1%, so that it is premature to eliminate 0.4 + 2 + 2 mol/L at this stage.

#### 3.2.1. CO$_2$ Desorption of with Solid Acid Catalysts with Temperature Programming

Figure 4 plotted catalytic CO$_2$ desorption profiles of 0.4 + 2 + 2~0.5 + 2 + 2 mol/L MEA-EAE-DEEA with temperature programming. Four commercial solid acid catalysts were selected as: H$\beta$, H-mordenite, HND-8, and HND-580. These catalysts had been implemented onto MEA-EAE-AMP and MEA-BEA-DEEA tri-solvents, and verified to be effective [11,13,14]. Therefore, they were selected for MEA-EAE-DEEA herein.

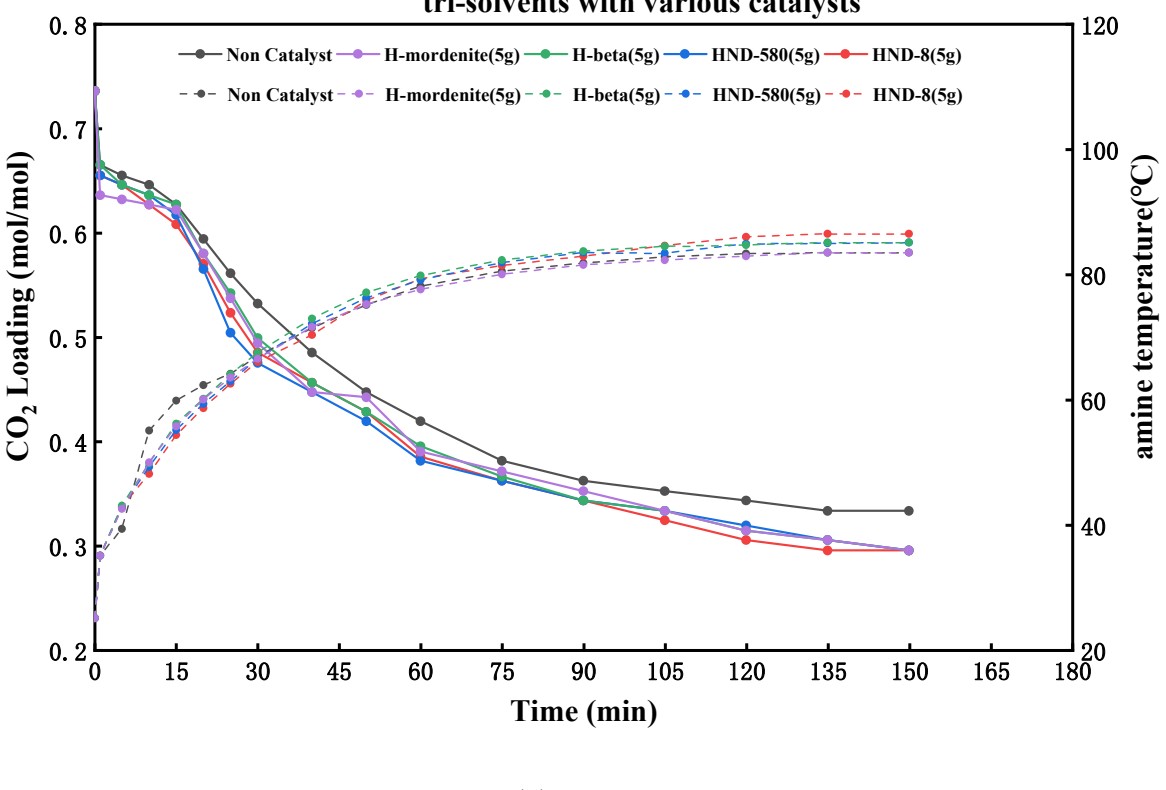

(**a**)

**Figure 4.** *Cont.*

**(b)**

**Figure 4.** The $CO_2$ desorption profiles of tri-solvents MEA-EAE-DEEA with several solid acid catalysts under temperature programming at (**a**) 0.4 + 2 + 2 mol/L and (**b**) 0.5 + 2 + 2 mol/L.

Figure 5 plotted the heat duties (HD) for catalytic desorption of MEA-EAE-DEEA solvents. The HD of catalytic desorption were smaller than non-catalytic counterparts at first 60 min, with an order as: non-catalyst > Hβ > H-mordenite > HND-8 > HND-580, and the smaller HD reflected the better desorption performance. The order was same for both 0.4 + 2 + 2 and 0.5 + 2 + 2 mol/L. For the same type of catalyst, the HD of 0.5 + 2 + 2 mol/L MEA-EAE-DEEA was smaller than that of 0.4 + 2 + 2 mol/L. This trend was the same as in Section 3.1.1. that the MEA-EAE-DEEA at 0.5 + 2 + 2 mol/L was the optimized condition at the range of 0.1 + 2 + 2~0.5 + 2 + 2 mol/L. The HD of temperature programming was smaller than direct heating since there is less heat loss based on inadequate heat input $Q_{in}$ than direct heating [14]. The smaller $Q_{in}$ resulted in smaller heat duty HD. Among 4 catalysts, the minimum HD was 477.5 kJ/mol for 0.5 + 2 + 2 mol/L MEA-EAE-DEEA + HND-580 at 30 min, which is below 500 kJ/mol compared to Figure 2 with direct heating.

As a result, this study discovered another energy-efficient combination that the minimum HD was located at the edge concentration of 0.5 + 2 + 2 mol/L. Based on other publications [11–15], the optimized blending ratios were different from the edge condition among 0.1 + 2 + 2~0.5 + 2 + 2 mol/L. For MEA-BEA-AMP and MEA-BEA-DEEA, the optimized mixing concentration of the tri-solvent was 0.3 + 2 + 2 mol/L [11,14,15]. For MEA-EAE-AMP, the optimized blending ratio was 0.2/2/2, different from edge of 0.5 + 2 + 2 [13].

Under those cases, it was necessary to analyze and compare the HD of 0.x + 2 + 2 mol/L + solid catalysts vs. max $C_A$ of 0.5 + 2 + 2 mol/L with solid acid catalysts carefully. Such comparison would not only find out the most energy-efficient combo, but also evaluate the "competition effect" of optimized blending ratio with catalysis vs. max $C_A$ with optimized catalysis. The dominant factor raised research interests, and it needs to be studied case by

case. For MEA-BEA-DEEA, 0.3 + 2 + 2 with catalyst was better than 0.5 + 2 + 2 in most cases [14]. For the case of MEA-EAE-DEEA, the effects were consistent toward 0.5 + 2 + 2 mol/L, which is quite convenient and applicable for the industrial application of solvents and catalysts.

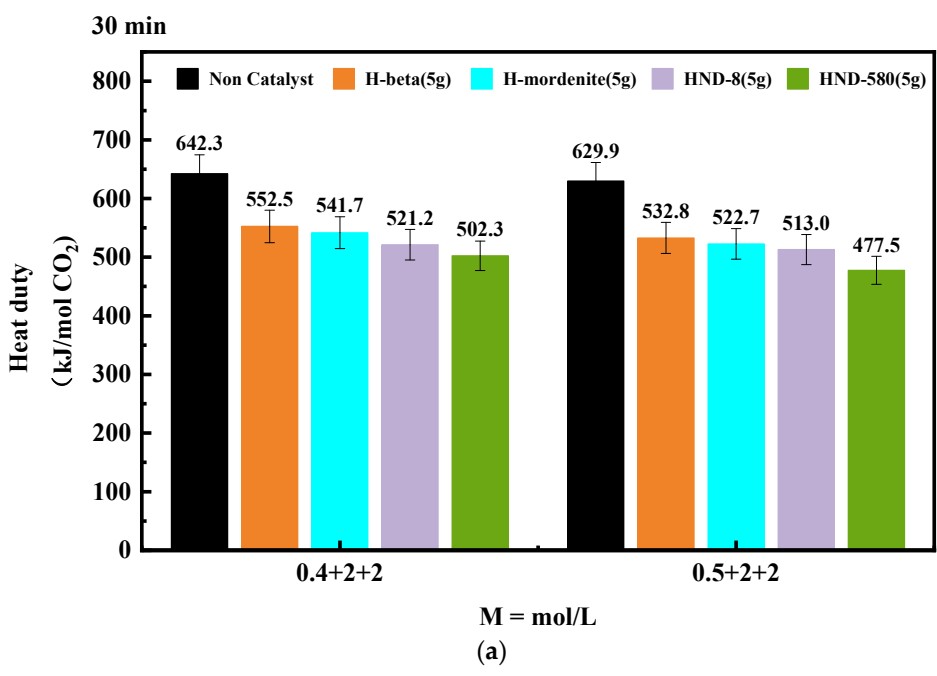

(**a**)

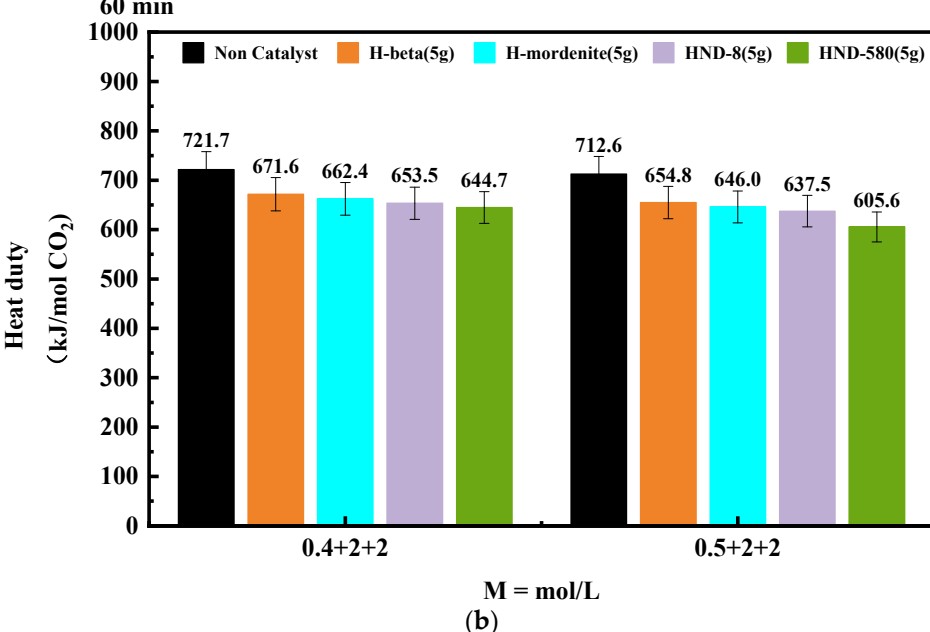

(**b**)

**Figure 5.** The heat duties of MEA-EAE-DEEA with various catalysts at (**a**) 30 min and (**b**) 60 min.

Optimized blending ratio + catalysis vs. Optimized catalysis + max $C_A$.

Furthermore, Figure 6 plotted the average desorption rates ADR at 30 and 60 min. The order of ADR was exactly the opposite of HD, which is: non-catalyst < Hβ < H-mordenite < HND-8 < HND-580, the higher ADR reflected the better desorption performance. These phenomena were reliable since better catalysts resulted in faster desorption rates and lower heat duty.

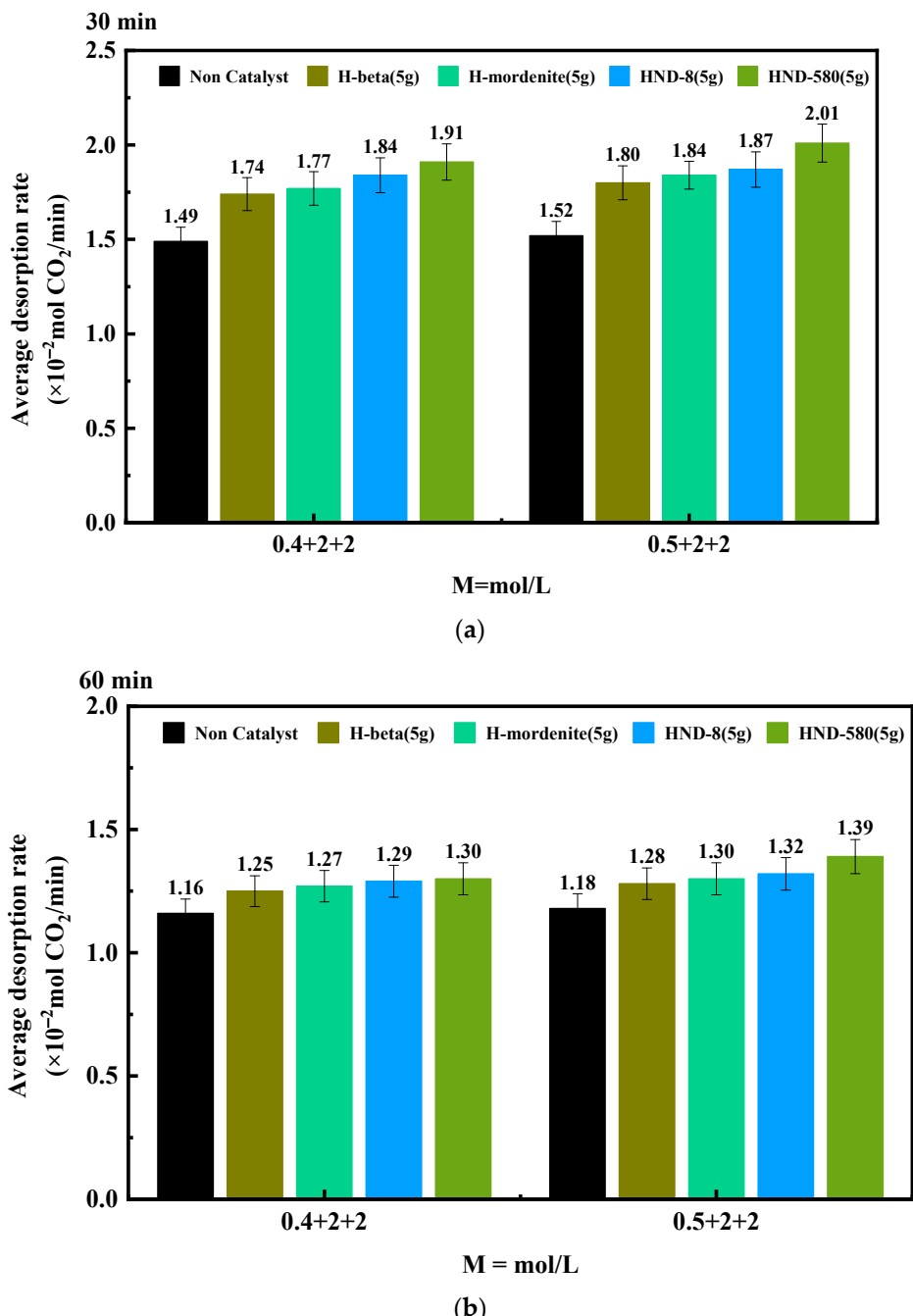

**Figure 6.** The average desorption rates (ARD) of tri-solvents MEA + EAE + DEEA with catalysts at (**a**) 30 min and (**b**) 60 min.

### 3.2.2. Compare This Study with MEA-EAE-AMP and MEA-BEA-DEEA Tri-Solvents with Solid Acid Catalysts

A previous study reported MEA-EAE-AMP blends with 5 g solid acid catalysts, such as: blended $\gamma$-$Al_2O_3$/HZSM-5 = 2:1, H$\beta$, H-Mordenite, HND-580, and HND-8 [12]. The HD plot of 30 min was studied. At $C_A$ of MEA-EAE-AMP was compared, the heat duty ranges 682.7~735.0 kJ/mol [12], much higher than MEA-EAE-DEEA of 477.5~532.8 kJ/mol. If the optimized condition of 0.2 + 2 + 2 mol/L MEA-EAE-AMP with various catalysts were compared, the heat duty ranges 679.1~723.3 kJ/mol [12], which is still higher. These results indicated the MEA-EAE-DEEA was a better tri-solvent than MEA-EAE-AMP at 0.1 + 2 + 2~0.5 + 2 + 2 mol/L with four commercial catalysts. Compared with the most energy-efficient combination of MEA-EAE-AMP at 0.2 + 1 + 3 mol/L, the heat duty was

589.3 kJ/mol with HND-8 and 594.3 kJ/mol for HND-580 [12]. The MEA-EAE-DEEA was still better.

On the other hand, the order of heat duty of catalytic MEA-EAE-AMP desorption was: non-catalyst > Hβ > H-Mordenite > HND-580 > HND-8. The smaller HD reflected better $CO_2$ desorption performance [12]. The HD and catalytic effect of HND-580 were only second to HND-8 but better than the other catalysts. The order was slightly different in this study, which is non-catalyst > Hβ > H-Mordenite > HND-8 > HND-580. The HND-8 was the best catalyst for MEA-EAE-AMP while HND-580 was the best for MEA-EAE-DEEA.

Based on Table 3 of acidity strength, pore diameter, pore volume, and surface area, the HND-8 has better acidic strength than HND-580, while the other properties were comparable [12]. From the heat duty analysis, the HD of HND-8 were very close to HND-580 under the same amine blends. From Figure 4, the desorption curve of HND-8 was worse and comparable than HND-580 at 0–120 min, but getting better at 120–150 min. The HND-8 has super performance at $\alpha_{lean}$ region < 0.35 mol/mol because of its strong acidity strength. Finally, the intrinsic structure-activity correlations required further study.

**Table 3.** Main properties of HND-8 and HND-580 from commercial labels [12].

| Parameters of Solid Acid Catalysts | Catalyst | |
|---|---|---|
| | HND-8 | HND-580 |
| Acidity by strength (mmol/g) | 24.75 | ≥4.95 |
| Surface area (m$^2$/g) | >20 | ≥20 |
| Wet apparent density (g/mL) | 0.75–0.85 | 0.55–0.65 |
| Wet true density (g/mL) | 1.18–1.28 | 1.18–1.28 |
| Average pore diameter (nm) | ≥15 | ≥15 |
| Pore volume (cm$^3$/g) | 0.2–0.4 | 0.2–0.45 |
| Particle size between 0.315–1.25 mm (%) | >90 | ≥90 |
| Water content (%) | ≤3 (dry), 50–57 (wet) | ≤3 |
| Maximum operating temperature (°C) | 150 | 170 |

Furthermore, MEA-EAE-DEEA was compared with MEA-BEA-DEEA [14]. The minimized heat duty was 477.5 kJ/mol for 0.5 + 2 + 2 mol/L MEA-EAE-DEEA + HND-580. The minimized HD of MEA-BEA-DEEA + HND-8 was 279.8 kJ/mol of 0.3 + 2 + 2 mol/L and 250.7 kJ/mol of 0.5 + 2 + 2 mol/L under the same operation conditions. The reason was that EAE has worse desorption performance but better absorption performance than BEA, which was verified repeatedly [12,13,24]. However, MEA-EAE-DEEA has a different operation region ($\alpha_{lean} \sim \alpha_{rich}$), as 0.30~0.72 mol/mol which was higher than that of MEA-BEA-DEEA 0.25~0.68 mol/mol [14]. The different operation condition was due to the secondary amine EAE owning higher cyclic capacity and absorption parameters than BEA [24]. However, the energy-efficient combination was also 0.5 + 2 + 2 mol/L + HND-8/HND-580 for both MEA-EAE-DEEA and MEA-BEA-DEEA, reflecting max $C_A$ with optimized catalysis [14].

### 3.3. The Optimized Mixture of MEA-EAE-DEEA with Catalysts Based on DF

The heat duty (HD) was a very important parameter indeed to evaluate energy efficient combinations. On the other hand, the desorption factor (DF) was a more comprehensive factor with cyclic capacity (CC) and average desorption rates (ADR) considered. The DF of 30 min were calculated and grouped in Table 4, and the 0.5 + 2 + 2 mol/L MEA-EAE-AMP with HND-580 was the best among the rest. After analysis of HD and DF, 0.5 + 2 + 2 mol/L MEA-EAE-AMP with HND-580 was the most energy-saving combo of $CO_2$ desorption among the rest sets.

**Table 4.** Desorption Factor of tri-solvent MEA-EAE-DEEA under Temperature Programming at 30 min.

| MEA-EAE-DEEA (mol/L) | Desorption Factor $(10^{-3}$ mol $CO_2)^3/L^2$ kJ min | | | | |
|---|---|---|---|---|---|
| | Non-Catalyst | H-Beta | H-Mordenite | HND-8 | HND-580 |
| 0.4 + 2 + 2 | 0.0209 | 0.0348 | 0.0348 | 0.0390 | 0.0416 |
| 0.5 + 2 + 2 | 0.0221 | 0.0366 | 0.0387 | 0.0410 | 0.0508 |

Finally, if we compare the HD and DF of several optimized combinations of various tri-solvent with catalysts, some clues were discovered. Table 5 reported three tri-solvents + catalysts: MEA-EAE-DEEA, MEA-EAE-AMP, and MEA-BEA-DEEA at optimized catalysis with temperature programming methods for the 0–30 min period. From Table 5, the HND-8 was the best catalyst for the most tri-solvents except MEA-EAE-DEEA. Generally, the heat duties follow the order: MEA-EAE-AMP + HND-8 > MEA-EAE-DEEA + HND-580 > or ≈ MEA-EAE-DEEA + HND-8 > MEA-BEA-DEEA + HND-8. The smaller HD reflected better desorption performance. Consequently, the DF follows the reverse order: MEA-EAE-AMP + HND-8 < MEA-EAE-DEEA + HND-580 < or ≈ MEA-EAE-DEEA + HND-8 < MEA-BEA-DEEA + HND-8. The bigger DF reflected stronger comprehensive desorption performance, including larger average desorption rates (ADR) and cyclic capacity (CC).

**Table 5.** The Heat duty and desorption factors of various tri-solvents with 5 g solid acid catalysts at optimized catalysis with temperature programming at 30 min.

| Tri-Solvents | Concentration (mol/L) | Catalysts | Desorption Factor $(10^{-3}$ mol $CO_2)^3/L^2$ kJ min | Heat Duty kJ/mol | Ref |
|---|---|---|---|---|---|
| MEA-BEA-DEEA | 0.3 + 2 + 2 | HND-8 | 0.1419 | 279.8 | [14] |
| | 0.5 + 2 + 2 | HND-8 | 0.1974 | 250.7 | [14] |
| MEA-EAE-AMP | 0.2 + 2 + 2 | HND-8 | 0.0223 | 679.1 | [12] |
| | 0.5 + 2 + 2 | HND-8 | 0.0220 | 682.7 | [12] |
| | 0.2 + 1 + 3 | HND-8 | 0.0277 | 589.3 | [12] |
| MEA-EAE-DEEA | 0.4 + 2 + 2 | HND-580 | 0.0416 | 502.3 | This study |
| | | HND-8 | 0.0390 | 521.2 | This study |
| | 0.5 + 2 + 2 | HND-580 | 0.0508 | 477.5 | This study |
| | | HND-8 | 0.0410 | 513.0 | This study |

From Table 5, the catalytic desorption performance of MEA-EAE-DEEA + HND-580 was in the middle of MEA-EAE-AMP + HND-8 and MEA-BEA-DEEA + HND-8. This performance was reasonable since EAE has stronger absorption performance than BEA but worse desorption factors. The HD of MEA-EAE-DEEA was bigger than MEA-BEA-DEEA [14]. On the other hand, the tertiary amine DEEA has better desorption performance than AMP [24], so that the MEA-EAE-DEEA has a smaller HD than MEA-EAE-AMP. The advantage of MEA-EAE-DEEA over MEA-BEA-DEEA was its higher operation regions of $\alpha_{lean} \sim \alpha_{rich}$ in a bench-scale process. The operation line of optimized tri-solvent MEA-EAE-DEEA was 0.35~0.72 mol/mol for 4.5 mol/L $C_A$, while that of MEA-BEA-DEEA was 0.28~0.65 mol/mol for 4.3 mol/L $C_A$. The bigger $C_A$ and $(\alpha_{rich} - \alpha_{lean})$ resulted in a bigger cyclic capacity (CC) at the same liquid flow rate ($F_l$) for the steady-state process, which is beneficial for desorption performance as preparation [46]. Therefore, tri-solvents of 0.4 + 2 + 2~0.5 + 2 + 2 MEA-EAE-DEEA + HND-8 or HND-580 could be beneficial to the steady state process.

## 4. Experimental Process

*4.1. Chemicals, Amines, Solid Acid Catalysts, and $CO_2$ Loading Analysis*

The amines (MEA, EAE, and DEEA) were purchased from Sigma Aldrich (SL, USA). The $CO_2$ gas was purchased with 99% purity. The solid acid catalysts were purchased as well, such as blended $\gamma$-$Al_2O_3$/H-ZSM-5, H$\beta$, H-mordenite, HND-580, and HND-8. The

HCl (1.0 mol/L) was used for titration and the methyl orange was used as the indicator. The $CO_2$ loading of amine solvents was tested and calculated with a Chittick apparatus (NY, USA), based on the Association of Official Analytical Chem (ists (AOAC), with a 2.5 % error [47].

*4.2. Experimental Procedures for Catalytic $CO_2$ Desorption for Temperature Programming*

This $CO_2$ desorption process has also been reported by other studies [14,48], consisting of a recirculation process and an oil bath [32–35]. Each set of experiments used 500 mL tri-solvent, and it was pre-loaded with adequate $CO_2$ to reach Vapor–Liquid Equilibrium. The process was heated to reach 90 °C with an oil bath, and then the desorption process took place to release $CO_2$. The Energy cost (Q) and $CO_2$ loading ($\alpha$) were recoded along with the process. Similar to MEA-BEA-DEEA [14] and MEA-EAE-AMP [12], the MEA-EAE-DEEA were also highly energy efficient. This study adopted direct heating to find out energy saving $C_A$ within at a concentration of 0.1 + 2 + 2~0.5 + 2 + 2 mol/L, and then use the temperature programming (TP) method to analyze the effect of catalytic $CO_2$ desorption [12,14]. For direct heating, the oil bath was set to 98–103 °C [29], and placed with flasks containing fully loaded amine concentrations [33]. For the TP method, the initial temperature was set at 25 °C of the oil bath and it increased gradually to reach 85~90 °C [12]. The $CO_2$ desorption curves were plotted along with temperature profiles in the same figure [12,14].

Besides, the catalytic $CO_2$ desorption process under the Temperature Programming method was indicative of benefits at such aspects: (1) different $CO_2$ loading was recorded as $\alpha_{lean}$ at 70, 80, and 90 °C, which reflects the operation conditions. (2) inadequate heat input $Q_{in}$ reflected the realistic condition of desorber for industrial amine scrubbing plants [12,14].

**5. Conclusions**

This study studies catalytic $CO_2$ desorption performance of MEA-EAE-DEEA with 5 solid acid catalysts: blended solid $\gamma$-$Al_2O_3$/H-ZSM-5 = 2:1, H$\beta$, H-mordenite, HND-8, and HND-580.

(1) The 0.5 + 2 + 2 mol/L turned out to be the optimized amine concentration, the same as the edge condition. This is the first case that the optimized blending ratio of MEA/EAE was consistent with max $C_A$.

(2) With several solid acid catalysts tested, the 0.5 + 2 + 2 MEA-EAE-DEEA with HND-580 catalysts was the best candidate with the order of DF: HND-580 > HND-8 > H-mordenite > H$\beta$. The catalysis of HND-580 was very close to HND-8 for this study, while the catalysis of HND-8 was better than HND-580 in most cases in other publications [12,14].

(3) Compared with other published combinations of tri-solvent + solid catalysts, the order of DF was MEA-BEA-DEEA + HND-8 > MEA-EAE-DEEA + HND-580 $\approx$ MEA-EAE-DEEA + HND-8 > MEA-EAE-AMP + HND-8. MEA-EAE-DEEA contains poorer desorption performance than MEA-BEA-DEEA but is better than MEA-EAE-AMP. However, this combination has a relatively big cyclic capacity and a special operation region of 0.35–0.72 mol/mol, which is also applicable in a steady-state process. The tri-solvent of MEA-EAE-DEEA would be adopted in a bench-scale pilot plant for the $CO_2$ absorption-desorption process, with solid acid catalysts HND-580/8 installed into the desorber. The HD in the steady state process will be conducted in the future.

**Author Contributions:** Conceptualization, H.S. and J.J.; methodology, Y.G.; validation, J.P. and Y.G.; formal analysis, L.J.; investigation, S.L.; data curation, Y.G.; writing—original draft preparation, review, and editing, J.J.; supervision, S.L. All authors have read and agreed to the published version of the manuscript.

**Funding:** This study was funded by the Bureau of Huzhou Municipal Science and Technology, (2021ZD2043, 2021ZD2003), the Bureau of Shanghai Municipal Science and Technology, (23010503500), and China Denmark International Cooperation Project (2022YFE0115800).

**Data Availability Statement:** The data was listed in the tables and figures.

**Conflicts of Interest:** The authors declare no conflict of interest.

## Glossary

Nomenclature

| | |
|---|---|
| CC | Cyclic Capacity |
| H | Heat duty (kJ/mol $CO_2$) |
| $nCO_2$ | Amount of $CO_2$ production (mol) |
| P | Total system pressure (kPa) |
| $Q_{input}$ | Heat input (kJ) |
| $C_A$ | Concentration of solute A in the bulk liquid (k mol/m$^3$) (mol/L) |

Greek Symbols

| | |
|---|---|
| $\alpha$ | $CO_2$ loading (mol $CO_2$/mol amine) |
| $\alpha_{eq}$ | $CO_2$ loading of solution in equilibrium with $PCO_2$ (mol $CO_2$/mol amine) |

Abbreviation

| | |
|---|---|
| AMP | 2-amino-2-methyl-1-propanol |
| BEA | Butylethanol amine |
| DEA | Diethanol amine |
| DEEA | N, N-diethylethanolamine |
| EAE | 2-(ethylamino)ethanol |
| MEA | monoethanol amine |
| PZ | Piperazine |

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
