# Peer review of "Catalytic CO2 Desorption Study of Tri-Solvent MEA-EAE-DEEA with Five Solid Acid Catalysts"

_catalysts, doi:10.3390/catal13060975_

Round 1

Reviewer 1 Report

The manuscript catalysts-2354253, “Catalytic CO2 desorption study of tri-solvent MEA-EAE-DEEA with five solid acid catalysts”, has been investigated the catalyzed desorption process of carbon dioxide in tri-blended amine. The topic is interesting but the quality of the article and scientific issues is poor and I recommend that the paper should not be acceptable for the publication.

1.      “2.2. The mechanism of Catalytic CO2 desorption”: The mechanism just proposed by title. More description needed.

2.      Eqs. 2,3: Calculation of heat with electrometer did not described. Is this method accurate? How about cyclic capacity?

3.      The experimental procedure is vague. How much solvent was used in each experiment? The details of setup were unknown.

4.      The certainty analysis and error analysis are missing and accuracy of the results are doubtful.

5.      Line 163: The results are very close Fig. 1. How the authors were presented 3% experimental error?

6.      Why the authors just optimized MEA concentration? What about EAE and DEEA?

7.      MEA-BEA-DEEA blend was more efficient than the tri-solvent of MEA-EAE-DEEA? Why the authors were emphasis on this solvent? The loading range of both solvents are close.

8.      Conclusions: “This is the first case that the optimized CA with catalysis was consistent 364 with max CA with optimized catalysis.” Is this sentence useful? What is the application of this sentence for CO2 capture process?

line 119:"into solvents,, which"

line 128: "This work contains two sets" --->three sets

Author Response

Please check attach file.

The revised manuscript will be provided separately.

Reviewer 2 Report

The authors have tried to demonstrate the CO2 desorption performance of amine blends and solid acid catalyst blends. The topic is important and fits within the scope of this journal. This work, however, lacks merit and needs further improvement.

1.   The authors have converged on optimum blends based on studies that are almost comparable to data within error bars. It is extremely important to provide an average over several studies and use an error bar collected from the same experiments, not use any theoretical estimate.

2.    Please also provide more control experiments with (i) no catalyst, (ii) no amines, and (iii) no catalyst and amine.

3.    There is no discussion of the roles of catalysts. Please elaborate on those.

Needs minor improvement

Author Response

See attach file.  The revised Figures will be included in the manuscript.

Reviewer 3 Report

Manuscript ID: catalysts-2354253

In this manuscript, CO2 desorption experiments were performed with a new tri-solvent MEA-EAE, DEEA with 5 solid acid catalysts. Sets of experiments were conducted in a typical recirculation process using of both heating directly and temperature programming method within at selected temperature, temperature ranges to evaluate the key parameters including average desorption rate (ADR), heat duty (HD), and desorption factors (DF).

This manuscript is very interesting and provides useful information to the related applications; however, some minor issues exist, and the discussion should be improved. I suggest that this manuscript could be considered for publication after suggested minor revisions are carefully made. The following comments should be useful for authors in revising their manuscript.

Q1) Don’t use too much abbreviation in abstract since readers have difficult to concentrate, some places meaning abbreviation have not been expressed. For instance, we know about MEA in line 12, but better to include what that short term means in first place it appears. line 14- Hβ, HND line 22-(αlean ~ αrich) etc

Q2) Line 28-36, I would suggest authors to include bad outcomes of having increasing CO2 concentration above 400 ppm. Also discuss some literature papers that use temperature programme desorption (TPD) or pressure swing adsorption (PSA) that currently used by scientist to regenerate CO2 under different conditions and discuss their disadvantageous over your suggested regeneration process. Please refer following papers and cite them appropriately

1 https://doi.org/10.1039/C9EN01442J

2 https://doi.org/10.3390/jcs6060168

3 https://doi.org/10.3390/ma11112301

Q3) Use one unit system either mol/L or M see Line 55, and 62

Q4) Line 58, “table,[9] which reported” which table you mean? Is it available in the manuscript or in reference? if reference, percent it in your manuscript under permission, if relevant

Q5) Line 91, line 131 you mention about 4 solid acid catalysts, but in the abstract discuss about 5, so check

Q6) I believe 2.1 and 2.2 should move to the results and discussion section since it is important to explain your observed results

Q7) Starting line 128, “This work contains two sets of: (1)…. (2)….. and then you put third (3), so better remove # 3 here but keep the content.

Q8) Starting from line 146, Move the Schemes, Figures, Tables before the text discussion pertaining to them, for instance Figure 1 (a-e) before line 146, readers can follow the content easily. Same principle will apply in all Schemes, Figures, Tables throughout the manuscript.

Q9) Starting from line 146, discuss results of Figure 1 (a-e), explain theory behind that and then jump to Figure 2, and then Figure 3 in consecutive order. Seems like some are bit mess here

Q10) Figure 2, 3, remove X scale unit mol/L in every place keep only one or use M symbol define M=mol/L underneath, move (a) 30 min (b) 60 min caption inside the figure not underneath

Q11) Keep space between unit and number for instance line 320 5g

Q12) Include something (1-2 sentences) that you plan to do in the future or important of this results to someone who expecting to carry our further research in similar areas

Author Response

Please see attach file, and the revised Figure were provided in the manuscript file.

Round 2

Reviewer 1 Report

Unfortunately, the accuracy of the results and experiments are doubtful. The presented value for Heat duty in the best case is 550 kj/molCO2, equals 12.5 Mj/kg CO2. This value for 30 wt.% benchmark MEA is 3.5-4.

Author Response

Dear Reviewer:

   The question is very good.  This value for 30 wt.% benchmark MEA is 3.5-4 GJ/tCO2.  The calculation of HD was based on the steady state process, which is

    HD = Qinput/nCO2 = mCp ΔT / FG0(Xin - Xout). 

    where the heat input is provided by the hot source, and mass liquid flowrate of amine with Δ T of liquid temperature.  The CO2 production was calculated by the gas flow rate with different CO2 concentration Xin and Xout.    

  However, the calculation of HD was different for the batch process of lab-scale experimental process.

   HD = Qinput/nCO2 = E /CAV(αrich - αlean)

The details were provided in attach file.
